# A Multidimensional Evaluation of the Factors in the Animal Welfare Assessment Grid (AWAG) That Are Associated with, and Predictive of, Behaviour Disorders in Dogs

**DOI:** 10.3390/ani14040528

**Published:** 2024-02-06

**Authors:** Rachel Malkani, Sharmini Paramasivam, Sarah Wolfensohn

**Affiliations:** School of Veterinary Medicine, University of Surrey, Guildford GU2 7AL, UK; s.paramasivam@surrey.ac.uk (S.P.); s.wolfensohn@surrey.ac.uk (S.W.)

**Keywords:** dog welfare, dog behaviour, welfare assessment, AWAG

## Abstract

**Simple Summary:**

Behaviour problems can impact the wellbeing of dogs. In this study, the researchers aimed to investigate the individual factors that influence behavioural disorders in dogs and how these impact their welfare. Dog professionals used an online welfare assessment tool, the animal welfare assessment grid (AWAG), to score dogs with behaviour problems. The researchers found that dogs’ clinical assessment, mobility, eating and drinking, aggression towards the caregiver, aggression towards unfamiliar people, reaction to stressors, the frequency at which they encounter fears and anxieties, choice, control, and predictability, use of enrichment, social interactions, beahvioural and handling during assessment, and change in daily routine as a result of a procedure or management event were scored poorer in dogs with behavioural disorders compared to healthy dogs. In addition, the dogs’ aggression towards the caregiver, fears and anxieties frequency, and choice, control, and predictability may predict behaviour problems. We suggest that veterinary and animal welfare staff should consider these as important indicators of emotional health in dogs.

**Abstract:**

Behavioural disorders in dogs are common and have severe welfare consequences for dogs. This study aimed to assess the factors that are significant and predictive of behaviour problems in dogs using the animal welfare assessment grid (AWAG) to further understand what factors influence their welfare. 177 AWAG assessments were undertaken across 129 dogs that clinicians deemed to have a behavioural disorder. Wilcoxon rank-sum tests were used to assess the difference in scores between dogs with behaviour disorders and a cohort of healthy dogs (*n* = 117). This analysis showed that all physical factors besides body condition, all procedural factors besides procedure pain, and all psychological, and environmental factors were significantly different between healthy dogs and dogs with behaviour disorders. Spearman rank correlation coefficient (RS) revealed several significant strong positive correlations including the procedural impact on the dog’s daily routine with aggression towards unfamiliar people and procedure pain, as well as other correlations between the dog’s behaviour during assessment with the frequency at which they encounter fears and anxieties, clinical assessment and procedure pain, and reaction to stressors and social interactions. These findings highlight the interdependent nature of the various influences of welfare. Logistic regression analysis identified that aggression towards the caregiver, fears and anxieties frequency, and choice, control, and predictability were all significant predictors of behaviour disorders. The findings have important implications for veterinary, behaviour, and animal welfare professionals as any changes across these factors may indicate poor welfare linked to emotional disorders in dogs.

## 1. Introduction

Behavioural disorders that stem from negative emotions such as fear, anxiety, and frustration can adversely impact the welfare of dogs. The prevalence of fear and anxiety in dogs is reported to be between 26% and 50% [1,2,3,4,5], and with an estimated 11 million dogs in the UK [6], which equates up to 5.5 million dogs potentially suffering from behaviour problems occurring from fear and anxiety.

Canine behavioural medicine is complex due to the uniqueness of each dog, in addition to the ongoing discussion regarding whether these behaviour problems should be considered as normal adaptive responses or dysfunctional responses. Moreover, the difficulty increases when attempting to classify these behaviours into distinct categories [7]. Therefore, assessing and treating affective disorders in dogs can pose significant challenges, necessitating a continuous effort. It is important for clinicians working in the field of behavioural medicine to understand how welfare is affected by emotional problems and the aetiology of behaviour.

Several factors can contribute to the development of behaviour disorders; these are often multifactorial and vary between dogs, and many of these interplay with each other. Negative past experiences or repeated trauma can lead to long-term fears and generalised anxiety in dogs, which will negatively affect their welfare. If the dog’s behaviour is unacceptable to the owner, it may result in inescapable physical or verbal punishment [8], further compromising the welfare of the dog.

Dogs may be predisposed to fearfulness and anxiety [9] and may possess certain traits that make them less capable of coping as effectively as other dogs. Behaviour traits such as excitability can underpin emotional responses. High levels of arousal can affect cognitive inhibition, resulting in anxiety and poor impulse control. Dogs are reported to have poor inhibitory control in a barrier detour task when they experience excessive excitement [10].

Emotional problems may also impact the cognitive abilities of dogs. Their ability to concentrate on tasks or stimuli can be compromised, resulting in impaired learning and problem-solving abilities [11]. However, low-to-moderate stress is also reported to improve cognitive function in other species [12]. There is also evidence to suggest that dogs are sensitive to changes in their caregiver’s affective state. Dogs’ working memory performance is shown to be correlated with changes in their owners’ self-reported stress levels; when the owners’ anxiety levels are manipulated in experimental conditions, dogs’ memory performance also changes in the same direction as their owners’ performance [13]. These studies highlight the complicated nature of the impact of stress and the interaction with emotions and cognition, which is likely to cause fluctuations in welfare state.

There may also be events that may minimally affect a dog’s wellbeing, but when experienced repeatedly, may result in cumulative stress. Examples of this may include a dog’s expectations consistently not being met. A single episode of frustration may have little impact on a dog’s wellbeing, but if this is prolonged or happens continually, the dog may experience an aversive emotional reaction, similar to fear and stress responses [14]. Another example may be a normal temporary reaction to a loud sound, but fear responses may both sensitise and generalise over time [15], and if a dog is exposed repeatedly to sounds that they cannot escape, they may develop strong fears and phobias. A phobia in animals involves a marked, persistent, and excessive fear of certain stimuli. The term phobia is derived from human psychiatry, where it describes an irrational fear. There is some debate over whether this term should be applied to animals; however, in the veterinary literature it generally describes an overly anxious reaction to specific stimuli [16].

One of the most common fears in dogs is loud noises and may occur in up to 49% of the population [5]; exaggerated responses can negatively impact welfare and attempts to escape the sounds can lead to self-harm and injury [16]. Noise phobias can also be indicative of a wider underlying anxiety problem and dogs with fear responses to sounds are reported to have other comorbid fears and anxieties [17] such as fear of crowds and other non-specific situational anxieties [2].

Abnormal behaviours in dogs can be indicative of negative affective states and poor welfare, and can often be misinterpreted by owners. Excessive emotional arousal is associated with behaviours exhibited outside of a normal context in a repeated and sustained manner [18]. These presentations have previously been categorised as obsessive, compulsive, or stereotypical behaviour and may be abnormal in duration, frequency and/or intensity, and they may be physically, cognitively or emotionally damaging to the dog [8]. However, as previously discussed, it should be considered if these behaviours are a normal adaptive response to the environment and situation the dog is in or if it is a pathological reaction.

The existing body of literature on the impact of chronic affective disorders on dog welfare is limited. However, considering the complexity of the assessment of canine behaviour disorders and their impact on welfare, it is of benefit to assess these using a standardised and comprehensive approach in order to monitor and improve quality of life. The animal welfare assessment grid (AWAG) is a tool that has been developed by the authors. It is a valid and reliable tool that quantifies welfare, is sensitive to change, and allows long-term monitoring over time (further details on development and validation can be seen in [19]. It assesses a range of factors that are evidence to influence welfare across four parameters (physical health, psychological health, the environment, and procedural and management events). This study aims to use to AWAG to explore which factors are significantly different in dogs with chronic behaviour problems compared to dogs’ clinicians have assessed to be physically and emotionally healthy, to investigate which factors are highly correlated with one another, and to examine which factors are predictive of emotional problems in dogs.

## 2. Methods

Veterinary surgeons, veterinary nurses, behaviourists, and animal welfare professionals were recruited to use the AWAG online tool via veterinary networks across the UK. A recruitment strategy involved study information posters being distributed to partner practices at the University of Surrey and in the Veterinary Times journal. Additionally, information about the project was shared extensively within the researcher’s professional networks. Recruitment posters were also shared on popular social networking platforms, including Facebook, Twitter, and LinkedIn. Interested individuals had the opportunity to share the recruitment link with their own networks, facilitating a wider dissemination of the project information.

Prior to accessing the AWAG site, users were required to sign a consent form, which outlined the purpose and ethics of the research. Clinicians also obtained consent from the owners of the dogs under assessment, ensuring that all parties involved were informed and agreed to participate. In cases where dogs were in shelter environments, a designated contact for the organisation was responsible for signing the consent form on behalf of all the dogs in their care.

Clinicians were provided with two user videos on using the AWAG. One which provides step-by-step guidelines on how to use the AWAG, and the other on how to access and interpret results. This ensures users have a standardised approach that promotes consistency when using the tool.

The AWAG was used to assess the welfare of dogs that were deemed by the assessing clinician to have a behavioural disorder. Assessments were undertaken from 23 June 2021 to 22 July 2023. In total, 129 dogs were assessed, and 177 assessments were undertaken across these dogs. Behaviour disorders could be categorised as either: abnormal or repetitive behaviour, anxiety disorders, phobias, separation-related problems, or other. Under the ‘other’ option, a free text box was available to write additional information about the condition. However, for the purpose of this study, any condition categorised under ‘behaviour condition’ was used in the analysis.

The AWAG involves users scoring each factor (Table 1) from 1 (indicating best welfare possible) to 10 (indicating worst welfare possible). Each score is accompanied by a written descriptor that is mutually exclusive from the other scores to make assessment as objective as possible (Appendix A) where the dogs are owned, the assessment is undertaken during a consultation with the other to obtain information about the dog’s history, psychological health and environment to accurately score the dog. Once the user has scored all factors, the AWAG calculates a cumulative welfare assessment score (CWAS) and a mean score for each parameter. The CWAS is the total area of the polygon which is generated by plotting the mean of each parameter across four axes on a radar chart. The CWAS is plotted over time to allow the user to see how welfare changes over time (Figure 1). The plot also shows how the dog compares to the ‘average healthy dog’ and plots against the mean, minimum, and maximum range from a cohort of dogs reported to be medically and emotionally healthy.

### Data Analysis

The AWAG produces mean scores for each parameter and a cumulative welfare assessment score (CWAS). These scores alongside the individual factor scores were used in statistical analyses. All analyses were performed using R Statistical Software (v4.0.1) [20].

Wilcoxon rank-sum tests were conducted to compare the factor scores between healthy (medically and emotionally) dogs and dogs with chronic pain. The scores of healthy dogs were obtained during validation and reliability studies of the AWAG [19] and healthy dogs scored up until 22 July 2023 (*n* = 117).

To assess which factors are highly correlated, a correlation matrix was calculated using the cor() function with Spearman rank correlation coefficient (RS) specified. The RS is a non-parametric statistical technique to assess the degree of linear correlation between two ordinal variables, which is often used in animal welfare science [21]. The correlation coefficients range from −1 to 1 indicating a stronger correlation the closer the value is to 1 or −1. Martin and Bateson (2021) classify correlations as very high (>0.9), high or strong (0.7–0.9), moderate (0.4–0.7), low (0.2–0.4) or weak (<0.2) [22] which will be used in this analysis.

To assess which factors in the physical, psychological, and environmental parameters may be predictive of chronic behaviour disorders, logistic regression models were fitted using the glm() function [23].

The model coefficients, *p*-values, and significance levels were extracted using the coef() and format.pval() functions. As several factors were either strongly correlated or were shown to be part of a poorer fitting model using likelihood ratio tests (LRTs) and AIC evaluation, 10 factors were removed from the model to avoid multicollinearity and to improve model fit. The final logistic regression model formula included:

logit(*p*) = *β*0 + *β*1Clinical.assessment + *β*2Aggression.towards.caregiver + *β*3Choice.control.and.predictability + *β*4Fears.and.anxieties.frequency + *β*5Procedure.pain

Following model section, variance inflation factor (VIF) was undertaken using the “car” package in R to examine multicollinearity among the predictor variables in the model. Multicollinearity can lead to unstable estimates and difficulties in interpreting the model results. Typically, VIF values of over five are reported to be highly correlated.

## 3. Results

### 3.1. Healthy vs. Behaviour Disorders

The mean score of healthy dogs was 4.94 and the scores ranged from 2.25 to 15. The mean score for dogs with behaviour disorders was 27.90 with a range of scores from 2.92 to 101.75 (Figure 2).

Every factor besides body condition score and procedure pain were significantly different between healthy dogs and dogs with chronic behaviour disorders (Table 2). The variation in scores for each factor can be seen in Figure 3.

### 3.2. Factor Correlation

Correlation coefficients revealed that there were several strong positive and negative correlations between various factors in dogs with behaviour conditions (Table 3) which were also statistically significant (*p* = <0.05) (Figure 4). Strong positive correlations (>0.7) between factors that are statistically significant include change in daily routine and aggression towards unfamiliar people (r = 0.78), behaviour during assessment and fears and anxieties frequency (r = 0.73), change in daily routine and procedure pain (r = 0.77), change in daily routine and eating and drinking (0.79), clinical assessment and procedure pain (r = 0.79), and reaction to stressors and social interactions (r = 0.71). A significant (*p* = <0.05) strong negative correlation was shown to be behaviour during assessment and eating and drinking (r = −0.76).

### 3.3. Factors That Are Predictors of Behaviour Disorders

Variance inflation factor (VIF) values indicated that there were no issues with high multicollinearity among the predictor variables. The VIF scores were clinical assessment = 1.19, aggression towards caregiver = 1.25, choice, control, and predictability = 1.13, fears and anxieties frequency = 1.42, and procedure pain = 1.11.

The logistic regression model revealed that clinical assessment (ß = 0.99, SE = 0.5, z = 1.99, *p* = 0.04), aggression towards caregiver (ß = 1.54, SE = 0.45, z = 3.47, *p* = <0.001), choice, control and predictability (ß = 0.94, SE = 0.37, z = 2.55, *p* = 0.01), and fears and anxieties frequency (ß = 0.81, SE = 0.17, z = 4.67, *p* = <0.001) were predictive of chronic behaviour disorders.

## 4. Discussion

This study aimed to assess which factors are significantly different in dogs with behavioural disorders compared to healthy dogs. Additionally, the research aimed to identify highly correlated factors and explore potential predictors of behavioural disorders in dogs using the quantifiable measures of the AWAG. Healthy dogs produced scores that were tightly clustered around the mean value, indicating a relatively consistent welfare assessment score. Conversely, dogs with behaviour problems displayed a much broader range of scores. The highest score among healthy dogs exceeded the mean by approximately 10 points, while in contrast, the highest score among dogs with behaviour issues was 74 points beyond the mean value. This demonstrates how variable the impact on welfare can be in dogs with behaviour disorders and also highlights the profound effect they can have on quality of life.

When examining the individual factors and how these differ between healthy dogs and dogs with behaviour disorders, all factors were significantly different besides body condition score and procedure pain. It is expected that dogs that are evaluated to be healthy are more likely to have good body condition due to the physical health risks associated with poor body condition [24,25]. There is a gap in the literature investigating associations between body condition and behaviour and these preliminary data may indicate that there is none as other factors are more important in influencing behaviour. However, frustration may occur in dogs with body conditions that impact their function and ability to carry out activities that are important to their welfare such as walking and playing, thus negatively impacting their emotional health. Regarding the lack of difference between the procedure pain score between healthy and dogs with behaviour issues, when being assessed, both cohorts of dogs were unlikely to be undergoing painful or invasive procedures due to their primary presentation.

The observation of significant differences in the remaining factors between healthy dogs and dogs with emotional problems is a notable finding. In terms of physical health, the poorer scores observed in clinical assessment and mobility and activity aligns with the existing literature, as behaviour problems are frequently associated with underlying medical conditions [26,27,28]. Eating and drinking can be influenced by stress and changes in emotional state [29,30] which may explain why this was poorer in dogs with behaviour disorders.

As predicted, all psychological parameters scored higher in dogs with behaviour issues compared to healthy dogs. Dogs are not distinctly inherently aggressive vs. non-aggressive; however, higher incidences of aggression are more likely to be present in dogs that have had traumatic experiences, poor socialisation and negative encounters as a puppy resulting in an inability to cope with stress as an adult in adulthood [31,32] in addition to genetic factors. Therefore, dogs that display aggression are more likely to be more fearful, anxious, and frustrated, resulting in behaviour problems. It is also important to acknowledge that aggression may be reinforced by positive outcomes, and this type of aggression may not occur in negative affective states.

Dogs assessed to have behaviour disorders scored poorer in their response to stressors (resilience) [33] and they encountered stressful stimuli more often than healthy dogs. Similar to dogs displaying aggression, the response to a perceived threat and the time it takes to recover may be influenced by genetics and their experiences as a young dog. However, it is unknown if these dogs are displaying behaviour problems because they are encountering specific fears frequently, or if the dogs have generally fearful and a variety of stimuli produce negative emotional responses.

Regarding the dogs’ environment, the ability to predict and control their surroundings and encounters is poorer in dogs with behaviour disorders. Increasing predictability and control is reported to ameliorate anxiety [34,35]. Predictable routines and outcomes create a sense of familiarity and security which can reduce negative responses to unexpected or unfamiliar stimuli. Having control over the environment means the dog can influence their surroundings, contributing to a reduced feeling of vulnerability, an emotion that is associated with negative valence.

Engagement in enrichment opportunities is also significantly different in dogs with behaviour problems compared to healthy dogs. The term enrichment is commonly used to describe animal care or management practices that help overcome deficits inherent in an animal’s environment or social life [36], but the definition can vary widely. In the AWAG tool, it is defined as “any addition to the dog’s environment that enhances the dog’s mental state”. These may be categorised as social, environmental, or mental enrichment; however, they are intrinsically interlinked, each impacting one another. Enrichment should be considered a fundamental inclusion to meet welfare needs and promote positive welfare, rather than an optional addition. A study investigating factors associated with fearfulness in dogs, a common initiating factor in behaviour problems, found that dogs engaging in activities and training more often display fewer signs of fearful behaviour [37]. This observation aligns with the AWAG data and suggests that engaging in enrichment is associated with better behavioural outcomes. In other species, enrichment is also shown to improve welfare by reducing stress, providing mental stimulation, allowing expression of natural behaviours and facilitating use of sensory abilities [38,39,40].

The other factor in the dog’s environment that is distinct between healthy dogs and dogs with behaviour disorders is their social interactions. The majority of healthy dogs have high-quality social interactions which are defined in the AWAG as direct engagement with people and/or dogs, depending on the individual’s needs which can involve activities such as play or positive training, or good-quality social interactions which involve indirect engagement such as walking or resting. Again, this is dependent on the individual dog’s needs and preferences. Domestic dogs are considered to be highly social and strong bonds exist between dogs and humans. These relationships are shown to be important to dogs, and a fundamental part of their wellbeing [41,42]. Maintaining and nurturing these bonds through positive engagement facilitates bonding and strengthen relationships to ensure the dog feels secure and safe in the human-dog dyad [43]. Dog–dog interactions, in particular play, can be important to a dog to fulfil their social needs [44]; however, there is wide individual variability between the social needs of dogs [45,46] and it was important to reflect this in the score descriptors to ensure that the individual’s preferences were taken into account. The AWAG scores reveal that dogs with behaviour problems are more likely to be socially isolated most days or more frequently with moderate to poor social interactions. This supports previous research in which dogs subjected to social restriction from people and other dogs showed signs indicative of stress and displayed increased aggressive responses [47].

Regarding procedural and management events, the dog’s behaviour during the assessment, their response to handling, and change in daily routine were different between healthy dogs and dogs with behaviour disorders. As anticipated, dogs with behavioural issues appear more fearful during the assessment and handling. Additionally, procedures are much more likely to take longer, impacting the dog’s daily routine. Dogs categorised as healthy are probably being assessed during a routine consultation such as a vaccination appointment, which are typically less than 20 min. A dog presenting with behaviour disorders normally has an extensive consultation lasting over two hours or if a dog with behavioural issues requires a veterinary procedure, this may take longer compared to an emotionally stable dog due to the potential difficulty in handling and need for sedation and psychoactive management.

The many correlations between factors in dogs with behavioural disorders highlights how the domains of welfare are not independent but interact and influence welfare positively and negatively. The analysis showed that change in daily routine (the amount of time taken out of the dog’s normal routine to undertake procures and management events) under the procedural parameter and also aggression toward unfamiliar people under the psychological parameter were positively correlated. This means dogs that had procedural and management events impact their daily routine at increasing frequency, also displayed more aggression towards unfamiliar people. This may suggest that dogs with behavioural disorders undergoing more invasive or lengthier treatment interventions have lower frustration tolerance, and as a result, may use repulsion towards veterinary staff as a means to avoid interaction. Thus, clinicians should consider if the procure is absolutely necessary as welfare-centric decision-making approach may suggest deferring treatment.

Conversely, dogs that display aggression towards unfamiliar people such as veterinary staff are going to be more difficult to handle and assess. Fear of the veterinary clinic is well-documented in dogs [48,49,50], and fear can often manifest as aggression, resulting in the need for longer modified procedures and/or use of sedation. In other environments such as the rehoming shelter, dogs that display aggression towards unfamiliar people will likely need behavioural management involving adapted handling and counterconditioning and desensitisation to strangers to change the emotional and behavioural response to people. In the change in daily routine factor, the longer the time taken for management and procedural events, the poorer the welfare is scored. However, in this situation, training and behavioural therapy could be improving the welfare of the dog, rather than compounding it. Therefore, when examining the change in daily routine factor, the context of the scenario should be considered rather than just assuming welfare is poorer because the dog’s routine has been impacted. This factor may be more valuable in a veterinary setting compared to other environments.

Change in daily routine was also positively correlated with procedure pain. Procedure pain is scored from no procedure required to extensive procedure resulting in severe long-term pain or complications. As painful procedures are typically going to be carried out in the veterinary environment, it is unsurprising that such interventions take longer and encroach into the dog’s daily time budget. Additionally, eating and drinking was also strongly positively correlated with change in routine—as consumption reduced, the time spent in the normal routine declined. Invasive veterinary procedures are carried out under general anaesthesia, which requires the dog to be starved prior to the treatment. Moreover, whilst hospitalised, the dog will unlikely be fed during any investigation.

Under the procedural parameter, behaviour during assessment was highly correlated with fears and anxiety frequencies which falls under the psychological parameter. Dogs that displayed body language of being uncomfortable and stressed during the AWAG assessment were more likely to encounter stressors (defined as a stimulus that the dog perceives to be frightening or threatening) more often. This observation holds true with generalised anxiety disorder (GAD) in dogs. Dogs with GAD exhibit signs of anxiety and fear triggered by a wide range of contexts, impairing quality of life and ability to function with daily life [51,52]. Acute physiological responses and emotional arousal resulting from fear and anxiety allow an animal to respond to a real or potential threat in their environment. These involve activation of the sympatho–adreno–medullary (SAM) axis and the hypothalamic–pituitary–adrenal (HPA) axis promoting catecholamine and glucocorticoid release [53]. Prolonged cortisol release caused by chronic stress can contribute to maladaptive physiological changes and the development of various disease which can manifest in protective emotional bias [18]. Excessive emotional arousal is also associated with abnormal behaviour and fear and aggression toward unfamiliar people or dogs [3]. Therefore, the results of these correlated factors support the existing evidence in that dogs display behaviours indicative of stress and protective emotional bias when they encounter a perceived or actual threatening stimulus at higher rates. In contrast, dogs that are calm, and actively engage with the assessor during the AWAG assessments, rarely encounter stressors.

The dogs’ response to stressors and their social interactions were found to be strongly correlated. In the factor descriptors, dogs that scored low displayed minimal signs of fear and anxiety and returned to normal quickly in response to a stressor. Dogs that took longer to recover had poorer social interactions. To our knowledge, there is extremely limited research examining resilience in dogs and the association of social interactions. In humans, engagement in social activity promotes higher levels of emotional stability and resilience [54] and it is suggested that positive interaction with dog caregivers through activities such as play may increase dog resilience and their ability to cope with stress [33]. When dogs have strong and secure attachments to their caregivers, they may develop better coping mechanisms to stressful events [55]. Moreover, being in proximity to an owner that they trust provides social support and security during a challenge. This has implications for veterinary practice, as often procures are undertaken in a separate area away from the owner. Being in close proximity to the owner may allow the dog to cope better and recover faster from a veterinary procedure that is likely to be stressful.

A final positive correlation that was found was between clinical assessment under the physical parameter and procedure pain. Clinical assessment is scored along a continuum from clinically healthy to extreme disease. Dogs with more severe behaviour disorders will score highly in their clinical assessment, and many dogs that present with behaviour problems, also have an associated medical condition [4,28]. Dogs with sound sensitivities are commonly reported to have chronic pain [27,56] and dogs with epilepsy can exhibit behavioural changes including anxiety [57,58]. Such dogs may require more advanced diagnostics and investigation which is likely to be more invasive and painful.

A strong negative correlation was identified between behaviour during assessment and eating and drinking. Eating and drinking behaviour was scored from best case eating and drinking as normal to anorexic or severe hunger or thirst at the worst case. This negative correlation suggests that as food and consumption reduced, dogs tended to be calmer during the AWAG assessment. Reduced appetite is a common clinical sign in dogs with medical problems [59] However, there is little in the peer-reviewed literature regarding the impact on appetite in dogs with behaviour disorders. Anorexia related to fear and anxiety in dogs with separation-related problems is most commonly cited when dogs are left alone [60]. This would suggest a positive correlation is more likely to occur over a negative. However, one possible explanation for this finding may be the result of psychoactive medication. Reduced appetite has been documented as a side effect in commonly used selective serotonin reuptake inhibitors (SSRIs) such as fluoxetine and tricyclic antidepressants (TCAs) such as clomipramine [61,62,63]. Dogs with behaviour problems treated with these medications may appear more calm and relaxed, with the associated concurrent loss of appetite.

Behaviour disorders in dogs are common reasons for relinquishment and euthanasia [64,65]. Predicting behaviour problems is of importance, as early intervention can prevent the escalation of issues and help protect welfare. Identifying behaviour problems is also important from a human safety perspective. Dog bites are an increasing public safety concern [66] and predicting potential problems can help mitigate the risk of bite incidents. Four factors (clinical assessment, aggression towards caregiver, choice, control and predictability, and fears and anxieties frequency) were found to be predictive of behaviour disorders in dogs. The growing evidence between behaviour disorders such as anxiety and comorbid medical problems suggests a strong connection between medical disorders and emotional problems. Medical conditions can have a direct or indirect impact on behaviour and Camps et al. [28] state that these conditions can be categorised into four primary groups: (1) problems that impact the perception of the animal’s environment such as blindness, (2) conditions that can disrupt neural pathways and sensory processing such as intracranial tumours [67,68] or problems that interfere with hormonal and neurological processes such as hypothyroidism and hyperadrenocorticism [69,70], (3) disorders that directly result in behaviour change such as pain, causing the animal to be protective and defensive [71], and (4) conditions or interventions that can prohibit the normal expression of behaviour such as tail-docking in dogs [72].

There is a growing body of evidence highlighting the interrelation between behaviour in dogs and associated medical problems [28,69]; therefore, poorer clinical assessment scores being a significant factor in dogs with behaviour disorders was unsurprising. Even after a medical problem has resolved, a dog may retain a learned behavioural response. For instance, if a dog previously experienced severe otitis, they might have developed avoidance behaviours or displayed aggression when approached, to prevent pain during handling. This association between pain and touch can persist long after the initial painful condition has been treated. [73,74].

Aggression towards the dog’s caregiver was also found to be predictive of behaviour disorders. Aggression in itself is a commonly cited behaviour problem and will likely have a strong emotional component. As aggression towards unfamiliar people was not found to be a significant factor in the model, only aggression towards the caregiver, this may suggest that the dog–owner dynamic may influence behavioural and emotional disorders in dogs. In humans, children with poorer-quality attachment with parents were found to be more aggressive compared to children with higher-quality attachments [75] and adolescents with mental health problems often experience difficulties with impulse control, leading to an increased likelihood of aggressive behaviour [76]. The child–caregiver bond and dog–caregiver bond are shown to be similar [77]. Therefore, it is reasonable to suggest that dogs with poorer-quality attachments with their owner may be more anxious and insecure in the relationship, and thus more likely to exhibit protective behaviours such as aggression. If the caregiver is unpredictable and inconsistent with their actions, or they use aversive training methods which can cause pain and anxiety, this can result in negative affective states and an increased likelihood of aggression [78,79,80].

Several neurotransmitters play a role in modulating aggressive behaviours. These include serotonin, noradrenaline, dopamine, acetylcholine (ACh), and gamma-aminobutyric acid (GABA). Low levels of serotonin can lead to increases in dopamine and noradrenaline, which can lower the threshold for aggression and increase impulsivity [81,82,83]. Dogs that display aggression have been reported to have lower levels of serotonin [84,85] which may be the result of pathologies or other abnormalities such as neurochemical imbalances. Conversely, mice subjected to chronic unpredictable stressors that included chronic stress are shown to have reduced serotonin levels [86]. Therefore, it is reasonable to infer that dogs in states of chronic stress will also have lower serotonin concentrations. Given serotonin’s role in modulating mood and emotion, dogs with affective disorders should be considered a higher risk to caregivers.

Dogs that may be predisposed to or have behaviour disorders may be less social towards other people; thereby encountering them less frequently, resulting in potentially fewer opportunities to display aggressive behaviours. Moreover, dogs will likely spend more time with their caregiver over unfamiliar people, which presents more opportunity for both positive and negative interactions to occur.

Reduced opportunities for choice, control, and predictability in a dog’s life was associated with behaviour problems. If a dog has no agency there is an increased susceptibility for distress. For example, if a dog is tied up or in a small, enclosed space, it cannot hide or escape a threat such as a loud noise, leading to chronic stress. Moreover, unpredictable environments can exacerbate any negative emotions, resulting in problematic behaviour.

Fears and anxieties frequency was also found to be a significant factor in the model, which is likely associated with a lack of control if a dog cannot escape a fearful situation. When fear and anxiety are experienced chronically or excessively, they are maladaptive and pathological, and considered an affective disorder that negatively impacts welfare. Dogs that were categorised as having a behaviour disorder experienced stimuli that caused a fearful or anxious response more often than dogs deemed as healthy. Frequent encounters with anxiety-provoking situations such as noisy, unpredictable environments or aversive interactions with conspecifics or people can lead to the development of (pathological) anxiety [87,88]. Regarding aggressive responses, dogs that are in anxiety-inducing environments often are more likely to be more sensitive to any stimulus they encounter, and sympathetic responses may cause them to react aggressively where a non-anxious dog would normally not respond. Anxiety and comorbid behavioural disorders such as reactive aggression (frustration and emotional dysregulation) are commonly associated with one another in the human literature. It is suggested that anxiety disorders may lower the threshold for various behavioural disorders in children [89], and the results of this study may provide similar evidence in dogs.

## 5. Limitations

The primary limitation of this study is that all behaviour disorders were grouped together in the analysis to obtain an adequate sample size. These various behaviour problem categories included: abnormal or repetitive behaviours, anxiety disorders, phobias, and separation-related problems. Within and between these categories, these disorders may have varying underlying emotions and motivators, so when assessed in isolation may have different factors associated with these. Moreover, the field of behavioural medicine lacks standardisation in the diagnosis of behavioural conditions. For example, separation anxiety is commonly viewed as a broad diagnosis, without considering underlying emotions such as fear, frustration, or panic. Dogs diagnosed as having ‘separation anxiety’ may display destructive behaviour when left alone but may be in a positive affective state.

Another limitation is related to the AWAG tool. Although each score has a written descriptor to improve objectivity, there may still be an element of clinician bias or users may have different interpretations of the wording. However, this should not impact the scores if the same user is scoring the dog over time.

### Future Research

Only few data were recorded about what type of procedures were undertaken. Compulsory data recording of the procedures and management events that were undertaken would allow a deeper exploration of how these interventions impact welfare. For example, evaluating the type of behaviour modification using the AWAG as an outcome measure would help measure the efficacy of various treatment plans.

## 6. Conclusions

The AWAG scores demonstrate the variation in welfare in dogs with behavioural disorders and how these are poorer compared to healthy dogs. The scores also highlight the individual factors than are influenced by behaviour problems. Analysis of the factor scores has revealed that clinical assessment, aggression towards the dog’s caregiver, choice, control and predictability, and fears and anxieties frequency are predictive of behavioural disorders in dogs. Additionally, the AWAG’s multidimensional approach, makes it possible to systematically identify areas where welfare may be compromised, and interventions can be prioritised to ensure prolonged suffering does not occur. This is a fundamental for ethical considerations across a range of dog settings.

## Figures and Tables

**Figure 1 animals-14-00528-f001:**
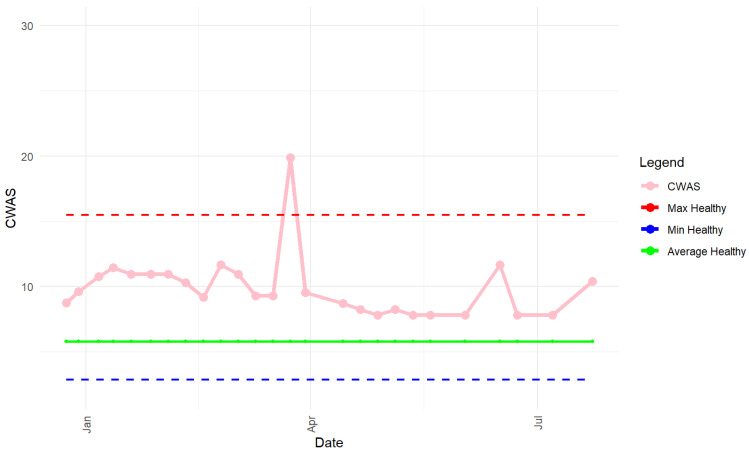
CWAS plot of a dog over a period of seven months. Lowest possible CWAS = 2, highest possible CWAS = 200.

**Figure 2 animals-14-00528-f002:**
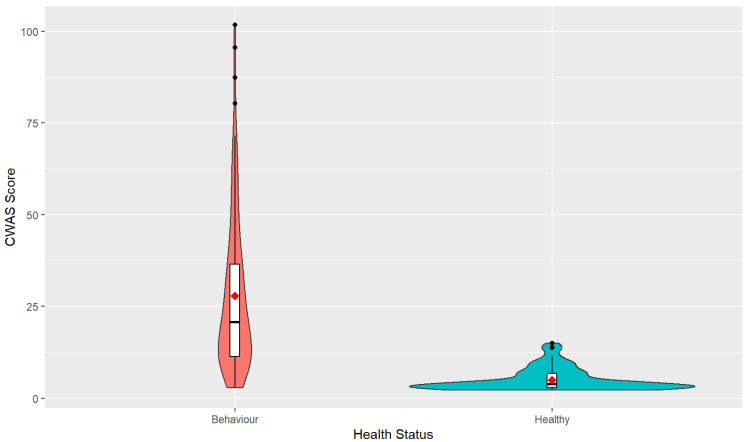
Violin plot of CWAS scores of dogs with behaviour disorders and healthy dogs. The violin plot represents the range and concentration of the scores with the widest part showing the highest frequency of scores. The centre of each violin plot shows a boxplot with the median line and first and third quartile data on either side. The mean scores are represented as the red point.

**Figure 3 animals-14-00528-f003:**
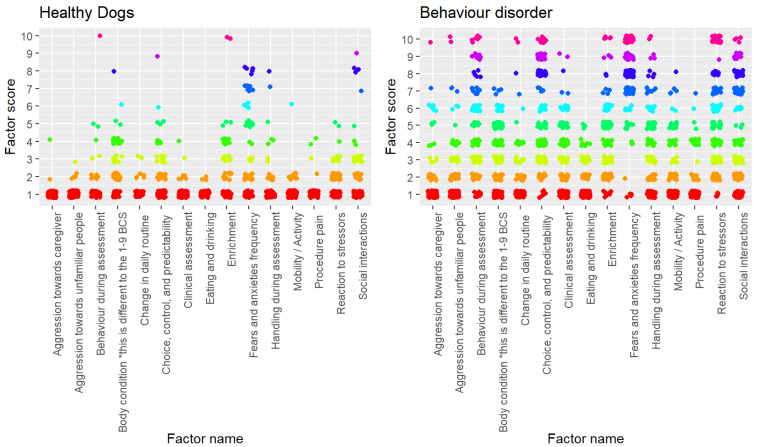
Plot of individual factor scores for each assessment (healthy dogs, *n* = 143; behaviour disorder dog, *n* = 177), colours pertain to each score (red = 1 best welfare possible, pink = 10 worst welfare possible). The * in body condition ensures users know the scoring is different from a standard 1–9 body condition scale.

**Figure 4 animals-14-00528-f004:**
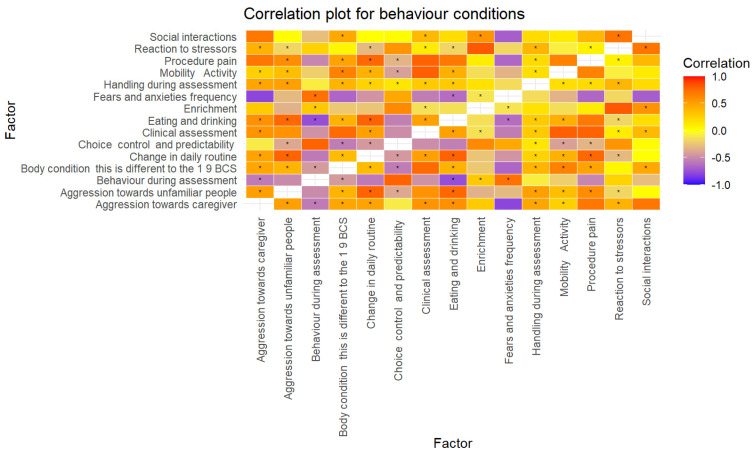
Correlation matrix of factors in dogs with behaviour disorders. * = *p* < 0.05.

**Table 1 animals-14-00528-t001:** List of parameters and the factors that are assessed in these categories.

Physical	Psychological	Environmental	Procedural
Mobility	Aggression towards caregiver	Choice, control, and predictability	Behaviour during assessment
Body condition	Aggression towards unfamiliar people	Enrichment	Change in daily routine
Clinical assessment	Fears and anxieties frequency	Social interactions	Handling during assessment
Eating and drinking	Reaction to stressors		Procedure pain

**Table 2 animals-14-00528-t002:** Results of Wilcoxon (rank-sum) tests between healthy dogs and dogs with behaviour disorders.

Factor	W Value	*p* Value
Aggression towards caregiver	8034	<0.001
Aggression towards unfamiliar people	8071	<0.001
Behaviour during assessment	6516	<0.001
Body condition score	8330.5	0.02019
Change in daily routine	2124.5	<0.001
Choice, control, and predictability	12,438	<0.001
Clinical assessment	337	<0.001
Eating and drinking	9182.5	<0.001
Enrichment	10,876	<0.001
Fears and anxieties frequency	936.5	<0.001
Handing during assessment	12,956	<0.001
Mobility/activity	450	<0.001
Procedure pain	7754.7	0.08476
Reaction to stressors	12,698	<0.001
Social interactions	9827.5	<0.001

**Table 3 animals-14-00528-t003:** Table of correlation coefficients (r) from Spearman rank correlation analysis.

	Aggression towards Caregiver	Aggression towards Unfamiliar People	Behaviour during Assessment	Body Condition	Change in Daily Routine	Choice Control and Predictability	Clinical Assessment	Eating and Drinking	Enrichment	Fears and Anxieties Frequency	Handling during Assessment	Procedure Pain	Mobility Activity	Reaction to Stressors	Social Interactions
Social interactions	0.707	0.011	−0.275	0.461	0	0.014	0.379	0.186	0.568	−0.689	0.2	0.386	0.114	0.711	1
Reaction to stressors	0.418	−0.175	0.246	0.043	−0.293	0.561	0.104	−0.175	0.821	−0.168	0.396	0.075	−0.071	1	0.711
Procedure pain	0.707	0.6	−0.518	0.493	0.768	−0.321	0.789	0.696	0.096	−0.654	0.193	1	0.646	0.075	0.386
Mobility Activity	0.257	0.354	−0.204	0.657	0.421	−0.4	0.8	0.429	−0.139	−0.325	0.179	0.646	1	−0.071	0.114
Handling during assessment	0.482	0.511	−0.075	0.407	0.243	0.139	0.289	0.286	0.161	−0.146	1	0.193	0.179	0.396	0.2
Fears and anxieties frequency	−0.789	−0.293	0.729	−0.657	−0.457	0.464	−0.557	−0.625	−0.114	1	−0.146	−0.654	−0.325	−0.168	−0.689
Enrichment	0.282	−0.329	0.293	−0.232	−0.246	0.618	−0.129	−0.136	1	−0.114	0.161	0.096	−0.139	0.821	0.568
Eating and drinking	0.586	0.764	−0.757	0.411	0.789	−0.546	0.496	1	−0.136	−0.625	0.286	0.696	0.429	−0.175	0.186
Clinical assessment	0.568	0.575	−0.468	0.754	0.514	−0.489	1	0.496	−0.129	−0.557	0.289	0.789	0.8	0.104	0.379
Choice control and predictability	−0.086	−0.368	0.786	−0.561	−0.436	1	−0.489	−0.546	0.618	0.464	0.139	−0.321	−0.4	0.561	0.014
Change in daily routine	0.493	0.782	−0.571	0.364	1	−0.436	0.514	0.789	−0.246	−0.457	0.243	0.768	0.421	−0.293	0
Body condition	0.479	0.4	−0.429	1	0.364	−0.561	0.754	0.411	−0.232	−0.657	0.407	0.493	0.657	0.043	0.461
Behaviour during assessment	−0.564	−0.5	1	−0.429	−0.571	0.786	−0.468	−0.757	0.293	0.729	−0.075	−0.518	−0.204	0.246	−0.275
Aggression towards unfamiliar people	0.514	1	−0.5	0.4	0.782	−0.368	0.575	0.764	−0.329	−0.293	0.511	0.6	0.354	−0.175	0.011
Aggression towards caregiver	1	0.514	−0.564	0.479	0.493	−0.086	0.568	0.586	0.282	−0.789	0.482	0.707	0.257	0.418	0.707

## Data Availability

The data presented in this study are available on request from the corresponding author.

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
