# Peer review of "A Multidimensional Evaluation of the Factors in the Animal Welfare Assessment Grid (AWAG) That Are Associated with, and Predictive of, Behaviour Disorders in Dogs"

_animals, 2024, doi:10.3390/ani14040528_

Round 1
Reviewer 1 Report
Comments and Suggestions for Authors
This paper is well-written and the need for this sort of assessment is known. However, the methodology is unclear to me, sorry.
In Table 1, it is written "Each score is accompanied by a written descriptor." Are these the descriptors in the Table? I did not find the text there to be behaviorally descriptive. How was "aggression" measured?" "Choice, control, predictability?" This reader cannot tell what you measured and how? THe same is true for "daily routine" and "handling;" what was it? What was done?
Maybe I am missing something, and clarification would be most welcome, thank you very much.
Author Response
Thank you so much for your comments.
I have uploaded the written descriptors as an appendix. Hopefully this helps with clarification on the methodology.

Reviewer 2 Report
Comments and Suggestions for Authors
Thank you for this contribution to the literature. The need for a strong welfare evaluation tool is necessary, particularly in the shelter world. The introduction was well done. I do have a few questions / comments about the paper:
1. The AWAG tool could be better described as far as what it is and why you selected it. It is not clear if the AWAG has been in use or if it was developed by you, etc. Perhaps a paragraph in the introduction to give the reader a better understanding of the tool.
2. Were the dogs only evaluated in stressful settings (vet office, shelter, etc.)? If so, would this impact their behaviors? A dog who does not exhibit 'aggressive' behaviors at home might show teeth or air snap when restrained at the vet office. How would this impact the study?
3. Was a history taken of the dogs? You mentioned negative past experiences play a role to fearful or anxious behaviors - are you making the assumption that fearful / anxious dogs have a traumatic past? Did the owners provide a history of where the dog was obtained, how long they have had the dog, etc?
4. In the introduction, you identify 3 aims it seems:
1. Which factors are different in dogs with chronic behavior problems vs those that are "healthy"
2. Which factors are highly correlated
3. Examine which factors are predictive of emotional problems
though the Discussion only addresses one "This study aimed to assess which factors are associated with behavior disorders in dogs and examine the state of welfare...." Perhaps that can be adjusted to be consistent in the paper.
5. I liked the use of charts and graphs to dispense the data.
Thank you again for your contribution.
Author Response
Thank you so much for your valuable feedback in improving this manuscript.
I have address these in the attached.
1. The AWAG tool could be better described as far as what it is and why you selected it. It is not clear if the AWAG has been in use or if it was developed by you, etc. Perhaps a paragraph in the introduction to give the reader a better understanding of the tool. - I have amended the introduction and cited the paper on development and validation of the AWAG tool.
2. Were the dogs only evaluated in stressful settings (vet office, shelter, etc.)? If so, would this impact their behaviors? A dog who does not exhibit 'aggressive' behaviors at home might show teeth or air snap when restrained at the vet office. How would this impact the study? I've attached a copy of the written descriptors as an appendix so is is clear how dogs are assessed. Behaviour is assessed by examining how often dogs are encountering fears, anxieties, and frustrations in their life in general, how the respond and recover to these, and also any aggression to people is assessed - these factors have been validated and are important to dog psychological health. Regarding behaviours in the vets or shelter - these are assessed under the procedural parameter which is a more 'at that moment' assessment. It assesses how the dog is behaving during the assessment to examine how they cope with procedures, including the assessment, as that can be a stressor in itself to some dogs. I hope this clarifies your question.
3. Was a history taken of the dogs? You mentioned negative past experiences play a role to fearful or anxious behaviors - are you making the assumption that fearful / anxious dogs have a traumatic past? Did the owners provide a history of where the dog was obtained, how long they have had the dog, etc? - this has been added to the methodology "Where the dogs are owned, the assessment is undertaken during a consultation with the other to obtain information about the dog’s history, psychological health and environment to accurately score the dog."
4. In the introduction, you identify 3 aims it seems:
1. Which factors are different in dogs with chronic behavior problems vs those that are "healthy"
2. Which factors are highly correlated
3. Examine which factors are predictive of emotional problems
though the Discussion only addresses one "This study aimed to assess which factors are associated with behavior disorders in dogs and examine the state of welfare...." Perhaps that can be adjusted to be consistent in the paper. - Have changed to "This study aimed to assess which factors are significantly different in dogs with behavioural disorders compared to healthy dogs. Additionally, the research aimed to identify highly correlated factors and explore potential predictors of behavioral disorders in dogs "
Reviewer 3 Report
Comments and Suggestions for Authors
Summary:
The study aimed to explore factors associated with behavioural disorders in dogs and the impact on their welfare. Utilising the Animal Welfare Assessment Grid (AWAG), authors assessed various factors across physical health, psychological health, the environment, and management strategies. The study included a sample of 129 dogs (177 assessments in total), with participation through a self-selected sampling method. Key findings included variability in welfare assessment scores between healthy dogs and those with behavioural issues, the influence of physical and psychological factors on behaviour, and the identification of predictive factors for emotional problems in dogs. Results will be of interest to canine professionals and dog owning members of the public.
Strengths:
· Validated Assessment Tool: The use of AWAG provided a standardised and holistic approach to assessing canine welfare which has been used in a range of research across different populations.
· Multifactorial Analysis: The paper recognised and addresses the complexity of behavioural disorders, considering multiple factors like past experience, genetic predispositions, and the dog's environment.
· Correlation and Predictive Analysis: The use of statistical tools to identify correlations and predictive factors adds an applied element to findings.
Weaknesses:
· Potential Bias in AWAG Tool: Despite standardisation, the AWAG tool's reliance on subjective assessments by clinicians could introduce bias.
· Grouping of Behavioural Disorders: The study grouped various behavioural disorders together, which might have masked differences between specific disorders.
· Lack of Longitudinal Data: The study, primarily cross-sectional, might not adequately capture the long-term impacts of behavioural disorders and interventions.
· Limited Environmental and Genetic Data: While the study acknowledged the role of environment and genetics, it lacked detailed exploration of these factors.
Potential Improvements:
Overall, the paper contributes valuable insights into the complex nature of canine behavioural disorders and their impact on welfare, using a comprehensive assessment tool and a robust sample size. The authors do acknowledge these potential limitations in the discussion section. However, the wider impact of these limitations on results and future studies/usability could be expanded. The methods section could be enhanced by including more reference to previous AWAG validation work (e.g. line 121).
Author Response
Thank you so much for your valuable feedback.
I have added this into the introduction "The Animal Welfare Assessment Grid (AWAG) is a tool that has been developed by the authors. It is a valid and reliable tool that quantifies welfare, is sensitive to change, and allows long-term monitoring over time (further details on development and validation can be seen in Malkani, 2022).
I have added this to the conclusion - "The AWAG's multidimensional approach, makes it possible to systematically identify areas where welfare may be compromised, and interventions can be prioritised to ensure prolonged suffering does not occur. This is a fundamental for ethical considerations across a range of dog settings."
I have also uploaded the written descriptors of each factor so it is clear how each dog is scored.